# Association between Physical Activity and Quality of Life in Colorectal Cancer Patients with Postoperative Defecatory Dysfunction: A Preliminary Survey

**DOI:** 10.3390/healthcare12141444

**Published:** 2024-07-19

**Authors:** Hiromi Nakagawa, Sho Hatanaka, Yoshimi Kato, Shinobu Matsumoto, Kiyoji Tanaka, Hiroyuki Sasai

**Affiliations:** 1Graduate School of Medicine, Gifu University, Gifu 501-1194, Japan; 2Research Team for Promoting Independence and Mental Health, Tokyo Metropolitan Institute for Geriatrics and Gerontology, Tokyo 173-0015, Japan; hatanaka@tmig.or.jp (S.H.); sasai@tmig.or.jp (H.S.); 3Uji-Tokushukai Medical Center, Kyoto 611-0041, Japan; y.kato@ujitoku.or.jp; 4Medical Research Institute Kitano Hospital, Osaka 530-8480, Japan; s-matsumoto@kitano-hp.or.jp; 5Faculty of Health and Sport Sciences, University of Tsukuba, Tsukuba 305-8577, Japan

**Keywords:** colorectal cancer, survivor, defecatory dysfunction, physical activity, sedentary behavior, quality of life

## Abstract

In this study, we aimed to explore the association between physical activity (PA) and quality of life (QoL) in colorectal cancer (CRC) patients with postoperative defecatory dysfunction. A survey using the European Organization for Research and Treatment of Cancer QLQ-30 and QLQ-29 was conducted among 62 adult outpatients with CRC at two cancer hospitals in Japan. PA and sedentary behavior were evaluated using the Global Physical Activity Questionnaire. Multiple regression analysis was performed, incorporating the QoL as the outcome, with the total PA and its three domains (occupational, transportation, and recreational) and sedentary time as exposures, while controlling for age, sex, and tumor location. The analyses revealed that patients engaged in PA ≥ 150 min/week (67.4 points; 95% confidence interval [CI]: 21.1, 113.8) and recreational PA ≥ 30 min/week (56.0 points; 95% CI: 2.3, 109.7) had significantly higher function scores. Conversely, sedentary time >8 h/day or occupational PA duration ≥30 min/week was associated with poor symptom and function scores. These findings highlight the importance of promoting recreational PA and reducing sedentary behavior to maintain and improve the QoL in CRC patients with defecatory dysfunction.

## 1. Introduction

Colorectal cancer (CRC) ranks as the second-most commonly occurring cancer worldwide [1]. According to the National Cancer Center in Japan, as of 2011, the 10-year survival rate had increased to 57.9% [2]. With the rapidly aging population, the number of individuals living with CRC is on the rise, highlighting the importance of maintaining and enhancing the quality of life (QoL). However, approximately 90% of the patients experience defecatory dysfunction [3] following anal-preserving surgery. Changes in the excretory pathway are associated with a higher prevalence of pain (odds ratio of 1.39) [4]. Moreover, a significantly higher prevalence of distress, depression, and anxiety has been reported in CRC patients compared to the general population [5]. Consequently, this leads to a substantial decline in the QoL in terms of both physical and mental health [6], posing an urgent challenge.

The American Society of Clinical Oncology (ASCO) guidelines [7] recommend 30 min of aerobic exercise and strength training per session, at least five times a week, totaling 150 min weekly, to alleviate the common side effects of cancer treatment and promote overall health. Adherence to these guidelines has been shown to significantly benefit cancer patients. For example, a study involving 112 CRC patients found that those who followed the American Cancer Society Nutrition and Physical Activity (PA) guidelines had better dietary habits and defecatory function [8]. Despite these benefits, less than half of the adults diagnosed with cancer, including 254 CRC patients, met the recommended PA guidelines [9]. However, those who met the guidelines experienced a notable reduction in fatigue. These findings emphasize the significance of regular PA in mitigating the adverse effects of cancer and enhancing patients’ overall well-being.

A systematic review of defecatory dysfunction and the QoL after rectal cancer surgery shows that poor bowel function affects the social and emotional functional domains of the QoL [10]. In addition, a high stool frequency, which can be inversely associated with the QoL, is independently associated with poor outcomes in CRC patients [11]. Thus, patients with defecatory dysfunction are a population whose QoL is significantly affected by their defecation-related conditions, while the association between PA and the QoL may have characteristics specific to this population. Many studies have examined the association between PA and the QoL in CRC patients, but research specifically on those with defecatory dysfunction remains limited [12,13,14]. In addition, because the severity of fecal incontinence is inversely associated with engagement in moderate-to-high levels of PA [15], defecatory dysfunction may affect the level of PA. Therefore, investigating the relationship between different levels of PA and the QoL may provide insights that account for the unique characteristics of PA in this population. Understanding the association between PA levels in different domains and the QoL in CRC patients with defecatory dysfunction helps identify the activity patterns of these patients and develop tailored healthcare strategies.

In this study, we aimed to investigate the association between PA and the QoL in CRC patients with postoperative defecatory dysfunction. The primary outcome was cancer- and CRC-specific function and symptoms, with exposure variables including the total PA, domain-specific PA, and sedentary behavior.

## 2. Materials and Methods

### 2.1. Study Design, Setting, and Patients

This multicenter cross-sectional study was conducted from October 2022 to March 2023 using anonymous self-administered questionnaires. This study was approved by the Institutional Review Board of Takarazuka University (no.: 2022-6; approval date: 4 July 2022) and the Medical Research Institute Kitano Hospital (no: P230201100; approval date: 15 February 2023). Participants attended outpatient clinics at two designated cancer hospitals in Japan, where physicians and certified wound, ostomy, and continence nurses randomly selected patients from the outpatient departments.

The study recruited 65 patients aged 20 and older (both male and female) who had postoperative defecation disorders due to colorectal cancer, with 62 patients providing complete responses to the questionnaire. They were divided into two groups: physically active (*n* = 30) and physically inactive (*n* = 32). The study included patients who had undergone CRC surgery within the past 20 years and had documented postoperative defecatory dysfunction, which included any difficulty in defecation. Exclusion criteria involved patients with local recurrence of CRC at the time of study registration; those undergoing radiation therapy or chemotherapy; those with mental functional disorders, with severe arrhythmias, or undergoing dialysis; and those with a history of or currently undergoing treatment for dementia. We also excluded participants with accidental bowel leakage. Written informed consent was obtained from all patients.

### 2.2. Measurement

#### 2.2.1. Primary Outcomes

The validated European Organization for Research and Treatment of Cancer (EORTC) Quality of Life questionnaires QLQ-30 [16] and QLQ-29 [17] for CRC were used for QoL assessment. The EORTC approved the use of the Japanese version of the EORTC QLQ-QoL survey. The QLQ-30 is a cancer-specific QoL questionnaire with 30 items, resulting in 15 aggregated scales as per the EORTC manual. These scales cover the overall QoL (global health status), functional domains (physical, role, emotional, cognitive, and social), and symptom domains (fatigue, nausea and vomiting, pain, dyspnea, insomnia, appetite loss, constipation, diarrhea, and financial difficulties). The scores in each domain were transformed into a range from 0 to 100. Higher scores on the overall QoL and functional scales indicated better conditions, whereas higher scores on the symptom scales indicated worse conditions. Total functional and symptom scale scores were calculated by summing the scores of the included domains (ranges 0–500 for total functional scales and 0–900 for total symptom scales).

The QLQ-29 [17] complements the QLQ-30 [16] as a CRC-specific functional questionnaire. We evaluated anxiety, weight, and body image as domains of the QLQ-29 functional scales, excluding sexual interest due to a substantial proportion of missing data. This missing may be owing to the cultural background of sexual interest among Japanese patients. Additionally, six symptom scales related to defecatory dysfunction were used: abdominal pain, rectal pain, abdominal bloating, flatulence, frequency of bowel movements, and fecal incontinence. Each domain score ranges from 0 to 100 points, similar to the QLQ-30 domains. Total functional and symptom scale scores were calculated by summing the domain scores, with ranges of 0–300 for total functional scales and 0–600 for total symptom scales. Higher functional scale scores indicated better conditions, while higher symptom scale scores indicated worse conditions.

#### 2.2.2. Exposure Variables

The exposure variables included the total physical activity (PA) as the primary exposure and domain-specific PA (occupational, transportation, and recreational) duration and sedentary time as secondary exposures. Occupational PA encompasses employment, volunteer work, part-time jobs, academic activities, household chores, caregiving, farming, fishing, job hunting, and other tasks.

The Global Physical Activity Questionnaire (GPAQ) [18], developed by the World Health Organization, is a validated tool for measuring PA across different countries. This questionnaire covers three domains (occupational, transportation, and recreational PA), providing a comprehensive overview of daily activity levels. Using the GPAQ to assess domain-specific PA is essential for designing effective strategies to improve the QoL in CRC patients.

In this study, PA levels were assessed using the GPAQ, which collects the PA frequency and duration in the aforementioned three domains. The GPAQ requires respondents to report PA lasting ≥10 min. Patients were divided into two groups based on the total PA duration using the ASCO-recommended cutoff (150 min per week). Domain-specific PA was categorized as ≥30 min and <30 min per day for activity levels and as ≥8 h and <8 h per day [19] for sedentary behavior.

#### 2.2.3. Other Characteristics

Data regarding patient characteristics were collected from medical records and questionnaires. These characteristics included sex, age, living status (alone or with someone), employment, postoperative weight change, body mass index, tumor locations, tumor–node–metastasis staging according to the Union for International Cancer Control, operative procedure, medical history, and defecatory dysfunction. Defecatory dysfunction encompassed multiple responses, such as fecal incontinence, difficulty defecating, frequent bowel movements, diarrhea, and constipation symptoms.

### 2.3. Statistical Analysis

The patient characteristics were compared between the “active” (≥150 min/week of total PA) and “inactive” (<150 min/week of total PA) groups using the Mann–Whitney U test for continuous variables and the chi-square test for nominal variables. Missing data were excluded list-wise from the analysis. Multiple regression analysis was conducted to investigate the association between the QoL and PA. The outcome variable was the QoL, and exposure variables included the total PA (<150 min/week, ≥150 min/week), as well as occupational, transportation, and recreational PA (<30 min/week or ≥30 min/week each) and sedentary time (<8 h/day, ≥8 h/day) [19]. All models were adjusted for sex [9,19], age [9,20], and tumor location [21,22,23,24]. Previous studies have adjusted for the tumor distance to the anal verge and the tumor location. In this study, tumor locations were identical to surgical sites. Given the limited sample size, only these three established confounding factors were included. Statistical analysis was performed using SPSS version 27 and R version 4.2.2, with significance set at *p* < 0.05.

## 3. Results

### 3.1. Patient Characteristics

Of the 65 CRC patients surveyed, 62 provided complete QLQ-30 data, yielding a 95.4% response rate. Table 1 shows the patient characteristics in terms of “active” (PA ≥ 150 min/week) or “inactive” status. The median weekly PA duration was 90.0 (IQR 0–210) min. In addition, 30 patients (48.4%) had a PA duration of 150 min or more per week. In terms of PA domains, 6 patients (9.7%) carried out occupational PA, 32 patients (51.6%) engaged in transportation activities, and 17 patients (27.4%) performed recreational activities, all for an average of ≥30 min. The occupations of the six individuals engaged in occupational PA were as follows: one automobile mechanic, one chef, and four office workers. The recreational activities of the 17 individuals included walking (*n* = 10, 58.8%), stationary cycling (*n* = 2, 11.8%), golf (*n* = 2, 11.8%), strength training (*n* = 2, 11.8%), and ballroom dancing (*n* = 1, 5.8%). However, 17 patients (27.4%) reported being sedentary for ≥8 h.

### 3.2. PA and QoL

Table 2 shows the comparison between the QLQ-30 and QLQ-29 scores in terms of PA status. In the QLQ-30, the physical, role, social, and total functional scale scores were significantly higher in the active group than in the inactive group. The overall QoL was also significantly higher in the active group (*p* < 0.05 for all comparisons). In the QLQ-29, the anxiety and weight scale scores, as well as the total functional scale scores, were significantly higher in the active group (*p* < 0.05) than in the inactive group. The symptom scales showed no significant differences between the groups. 

Table 3 presents partial regression coefficients and 95% confidence intervals (CIs), adjusted for sex, age, and tumor location. The total PA duration was the primary exposure variable, while the domain-specific PA (occupational, transportation, and recreational) duration and sedentary time were secondary exposures. Total PA ≥ 150 min/week was positively associated with QLQ-30 function scores (67.4; 95% CI: 21.1, 113.8), indicating a better QoL. Similarly, recreational PA duration ≥ 30 min/week was also positively associated with QLQ-30 function scores (56.0; 95% CI: 2.3, 109.7). Conversely, sedentary time ≥ 8 h/day was linked to higher QLQ-30 symptom scores (79.8; 95% CI: 10.5, 149.1) and lower function scores (−60.2; 95% CI: −113.2, −7.3), indicating poor QoL outcomes. Engaging in occupational activities for ≥30 min/week was associated with unfavorable QLQ-29 symptom scores (111.3; 95% CI: 43.0, 179.6) and function scores (34.6; 95% CI: 3.2, 66.1), suggesting a potential adverse association with the QoL in this domain.

## 4. Discussion

In this study, we examined the association between PA and the QoL in CRC patients with postoperative defecatory dysfunction. The results showed that total PA ≥ 150 min/week or recreational PA ≥ 30 min/week is associated with better functional scale scores. Active patients had significantly higher scores on the physical, role, and social subscales than their inactive counterparts. These findings suggest that higher levels of PA are associated with increased social activity, which is consistent with the results of Hirschey et al.’s study [25]. This indicates that recommending PA to CRC patients may prevent activity and functional decline. Previous research has demonstrated the benefits of exercise interventions in QoL domains, such as physical, role, and social functions [12], which were observed in this study as well.

When sedentary time exceeded 8 h per day, both symptom and functional scales were negatively impacted. When the occupational PA duration was ≥30 min per week, the symptom scale also reflected poorer outcomes. Cancer-related fatigue (CRF) significantly impacts the QoL, particularly physical function and the ability to perform daily activities [26]. This study suggests that inactive patients experience significantly worse anxiety than their active counterparts, indicating poorer mental health among inactive CRC patients. A previous study has reported that CRC patients who experience somatization and anxiety are at a higher risk of insufficient PA [27]. The reason for lower anxiety in active patients is that they can manage and cope with symptoms following CRC surgery, while maintaining their social and role functions. This aligns with our previous qualitative study [28] on defecation dysfunction in CRC patients, which identified categories such as “coping with defecation dysfunction” and “compromising with defecation dysfunctions”. There is a notable lack of research on the evaluation of the long-term effects of PA interventions on cancer-related anxiety [29].

The symptom and functional scales worsened when sedentary behavior exceeded 8 h per day. Inactive patients exhibited weight loss, leading to a significant difference in the weight sub-score of the QLQ-29 functional scales. Considering that 48% of CRC patients have sarcopenia [30], preventing weight loss due to muscle weakening associated with sedentary behavior is crucial. Prolonged sedentary behavior decreases cardiorespiratory function and functional status, exacerbating prolonged sitting during cancer treatment [31] and recovery phases, ultimately reducing the QoL. Interventions to promote habitual avoidance of sedentary behavior are needed, given the association with an improved QoL and the alleviation of CRF [32].

Our findings suggest that individuals with CRC experiencing postoperative defecatory dysfunction can potentially maintain a better QoL by reducing sedentary behavior and engaging in PA. This may help alleviate CRF and prevent physical function decline. Investigating the association between domain-specific PA and the QoL offers a practical way to consider behavioral prescriptions for individuals with CRC, highlighting its clinical significance.

This study has some limitations. First, the small sample size limits the generalizability of the findings. While an association between PA and the QoL was observed in CRC patients experiencing defecatory dysfunction, further large-scale studies are necessary to confirm these findings. Second, since this study was cross-sectional, causal relationships could not be inferred; therefore, further longitudinal studies are warranted. Third, the use of questionnaire surveys may have introduced recall bias, and self-reported PA may differ from device-based activity measurements. Future research should use activity monitors to analyze PA intensity and step data to reduce bias. Fourth, it remains unclear which specific types, durations, and intensities of PA are most strongly associated with a better QoL in CRC patients. Further research should investigate the effective duration and intensity of PA for maintaining and improving the QoL and develop intervention programs accordingly.

## 5. Conclusions

PA is associated with the QoL in CRC patients with defecatory dysfunction. Engaging in total PA for at least 150 min per week and recreational PA for at least 30 min per week is associated with better functional outcomes. In contrast, spending ≥8 h per day in sedentary activities or engaging in occupational PA for at least 30 min per week is associated with poorer functional outcomes and more severe symptoms. These findings underscore the importance of reducing sedentary behavior and promoting recreational PA to maintain and enhance the QoL in this patient population. Future research should validate these results through large-scale studies to overcome the limitations of this study’s small sample size. Additionally, using wearable activity trackers to assess the PA intensity and step count will help mitigate recall bias and yield more accurate measurements.

## Figures and Tables

**Table 1 healthcare-12-01444-t001:** Demographic characteristics of patients.

Item		Active (*n* = 30)	Inactive (*n* = 32)	*p* Value
Sex	Male	20 (66.7%)	22 (68.8%)	1.00
	Female	10 (33.3%)	10 (31.2%)	
Age	Years	66.8 (13.2)	65.5 (14.0)	0.72
Living status (alone or with someone)	Living alone	6 (20.0%)	5 (15.6%)	0.96
Employment status	Employed	9 (30.0%)	11 (34.4%)	0.92
Postoperative weight change	kg	0.1 ± 6.4	−2.5 ± 9.2	0.20
BMI	kg/m^2^	21.0 ± 3.5	22.1 ± 3.7	0.90
	<18.5 (underweight)	4 (13.3%)	7 (21.9%)	0.71
	>25.0 (overweight/obese)	6 (20.0%)	6 (18.8%)	
Tumor locations	Rectal	24 (80.0%)	27 (84.4%)	0.75
	Colon	6 (20.0%)	5 (15.6%)	
TNM-UICC stage	I	7 (23.3%)	5 (15.6%)	0.29
	II	5 (16.7%)	12 (37.5%)	
	III	12 (40.0%)	10 (31.3%)	
	IV	5 (16.7%)	2 (6.2%)	
	Unknown	1 (3.3%)	3 (9.4%)	
Operative procedure	LAR	14 (46.7%)	18 (56.3%)	0.36
	Colectomy	2 (6.6%)	9 (28.1%)	
	ISR	0 (0.0%)	1 (3.1%)	
	Others	14 (46.7%)	4 (12.5%)	
Medical history	Diabetes mellitus	9 (30.0%)	8 (25.0%)	0.78
	Hypertension	8 (26.7%)	13 (40.6%)	0.29
	Stroke	1 (3.3%)	4 (12.5%)	0.36
	Angina pectoris	3 (10.0%)	3 (9.4%)	1.00
	Dyslipidemia	5 (16.7%)	3 (9.4%)	0.47
	Respiratory illness	5 (13.3%)	4 (12.5%)	0.85
	History of fractures	5 (16.7%)	7 (21.9%)	0.75
Defecatory dysfunction	Fecal incontinence	12 (40.0%)	14 (43.8%)	0.80
	Evacuation difficulties	11 (36.7%)	14 (43.8%)	1.00
	Frequent stool	9 (30.0%)	9 (28.1%)	1.00
	Diarrhea	7 (23.3%)	10 (31.3%)	0.27
	Constipation	5 (16.7%)	5 (15.6%)	0.86

Data are presented as *n* (%) or the mean ± standard deviation. BMI, body mass index according to the World Health Organization classification; TNM-UICC, tumor–node–metastasis staging according to the Union for International Cancer Control; LAR, low anterior resection; ISR, intersphincteric resection.

**Table 2 healthcare-12-01444-t002:** QoL scores by physical activity status.

Item		Physical Activity		
		Active (*n* = 30)	Inactive (*n* = 32)	*p* Value
QLQ-30 (cancer-specific scale)				
Functional scores (the higher, the better)	Physical (scores 0–100)	86.0 (17.1) *	72.7 (21.6)	0.01
	Role (scores 0–100)	82.2 (22.7) *	60.4 (35.9)	<0.01
	Emotional (scores 0–100)	86.9 (15.6)	78.4 (20.3)	0.07
	Cognitive (scores 0–100)	77.8 (18.2)	73.4 (27.7)	0.47
	Social (scores 0–100)	84.4 (22.7) *	64.6 (32.2)	<0.01
	Total (scores 0–500)	417.4 (72.5) *	349.5 (113.7)	<0.01
Symptom scores (the lower, the better)	Fatigue (scores 0–100)	27.4 (19.5)	39.6 (28.8)	0.06
	Nausea and vomiting (scores 0–100)	3.3 (11.1)	6.3 (13.2)	0.35
	Pain (scores 0–100)	19.4 (21.5)	26.6 (27.1)	0.26
	Dyspnea (scores 0–100)	16.7 (19.1)	27.1 (32.2)	0.13
	Insomnia (scores 0–100)	33.3 (57.4)	37.5 (33.6)	0.73
	Appetite loss (scores 0–100)	22.2 (28.1)	21.9 (31.3)	0.96
	Constipation (scores 0–100)	7.8 (16.8)	12.5 (23.6)	0.37
	Diarrhea (scores 0–100)	12.2 (25.5)	20.8 (27.8)	0.21
	Financial difficulties (scores 0–100)	23.3 (27.9)	35.4 (36.8)	0.15
	Total (scores 0–900)	165.7 (120.0)	227.6 (147.4)	0.08
Overall QoL (scores 0–100)		62.2 (25.3) *	50.8 (18.4)	0.05
QLQ-29 (CRC-specific scale)				
Functional scores (the higher, the better)	Anxiety (scores 0–100)	71.3 (24.8) *	54.8 (31.7)	0.03
	Weight (scores 0–100)	81.6 (21.1) *	67.7 (29.2)	0.04
	Body image (scores 0–100)	69.0 (24.4)	63.1 (29.0)	0.40
	Total (scores 0–300)	221.8 (58.1) *	185.7 (77.0)	0.05
Symptom scores (the lower, the better)	Abdominal pain (scores 0–100)	13.8 (18.9)	12.9 (23.9)	0.87
	Buttock pain (scores 0–100)	19.5 (26.0)	31.2 (39.4)	0.19
	Bloating (scores 0–100)	18.4 (24.5)	19.4 (25.5)	0.88
	Flatulence (scores 0–100)	41.4 (27.7)	46.2 (26.8)	0.49
	Stool frequency (scores 0–100)	19.5 (30.2)	16.7 (22.8)	0.68
	Fecal incontinence (scores 0–100)	18.4 (22.9)	22.6 (23.4)	0.47
	Total (scores 0–600)	131.0 (93.0)	148.9 (98.7)	0.47

Data are presented as the mean and standard deviation. Groups are defined as a physical activity duration of over 150 min/week (active) and under 150 min/week (inactive). QLQ-30, European Organization for Research and Treatment of Cancer (EORTC) QLQ-30; QLQ-29, EORTC QLQ-29; QoL, quality of life; CRC, colorectal cancer. * Significantly (*p* < 0.05) better QoL scores.

**Table 3 healthcare-12-01444-t003:** Total and domain-specific physical activity and sedentary behavior for the QoL.

	Cancer-Specific Scale			CRC-Specific Scale	
	Overall QoL (QLQ-30) (Scores 0–100)	Symptom Scales (QLQ-30) (Scores 0–900)	Functional Scales (QLQ-30) (Scores 0–500)	Symptom Scales (QLQ-29) (Scores 0–600)	Functional Scales (QLQ-29) (Scores 0–300)
Total PA ≥ 150 min/week (ref. < 150 min/week)	10.8 (−0.2, 21.8)	−56.4 (−119.8, 7.0)	**67.4 (21.1, 113.8)**	−11.8 (−56.4, 32.8)	**34.6 (3.2, 66.1)**
Occupational PA ≥ 30 min/week (ref. < 30 min/week)	4.6 (−14.7, 23.9)	44.0 (−66.1, 154.1)	−0.3 (−84.8, 84.2)	**111.3 (43.0, 179.6)**	−24.4 (−78.8, 30.1)
Transportation PA ≥ 30 min/week (ref. < 30 min/week)	7.1 (−4.2, 18.3)	−54.4 (−118.1, 9.4)	31.2 (−18.0, 80.4)	−8.2 (−52.8, 36.5)	−0.7 (−33.6, 32.2)
Recreational PA ≥ 30 min/week (ref. < 30 min/week)	6.9 (−5.7, 19.5)	−65.4 (−136.3, 5.6)	**56.0 (2.3, 109.7)**	−12.5 (−62.3, 37.4)	17.9 (−18.5, 54.3)
Sedentary time ≥ 8 h/day (ref. < 8 h/day)	−1.4 (−14.0, 11.3)	**79.8 (10.5, 149.1)**	**−60.2 (−113.2, −7.3)**	9.7 (−40.7, 60.1)	−25.8 (−62.2, 10.6)

QLQ-30, European Organization for Research and Treatment of Cancer (EORTC) QLQ-30; QLQ-29, EORTC QLQ-29; 95% CI, 95% confidence interval; QoL, quality of life; PA, physical activity; CRC, colorectal cancer. All models were adjusted for sex, age, and tumor location. Bold values indicate statistical significance.

## Data Availability

The study data are available upon request from the corresponding author.

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
