# Peer review of "Association between Physical Activity and Quality of Life in Colorectal Cancer Patients with Postoperative Defecatory Dysfunction: A Preliminary Survey"

_healthcare, 2024, doi:10.3390/healthcare12141444_

Round 1

Reviewer 1 Report

Comments and Suggestions for Authors

1 Why you choose CRC patients with postoperative defecatory dysfunction? This was not mentioned in the introduction part.

Cite more scholarly references published in the past 2-3 years.

3 Can the small sample size provide sufficient statistical power?

Reviewer 2 Report

Comments and Suggestions for Authors

There are already several papers showing that physically active survivors of colorectal cancer have a better quality of life than survivors who were not physically active...please emphasise the importance of your work.

1.- What were the main occupational and recreational physical activities that  the patients performed?

2.- How would you classify the type of physical activity that the patients performed? aerobic or resitance or both?

3.- How many women participate in your study? your table 1 shows only the male population

4.- The association between physical activity and quality of life seemed to be stronger among women than among men (Eyl RE, Xie K, Koch-Gallenkamp L, Brenner H, Arndt V. Quality of life and physical activity in long-term (≥5 years post-diagnosis) colorectal cancer survivors - systematic review. Health Qual Life Outcomes. 2018;16(1):112. Published 2018 Jun 1. doi:10.1186/s12955-018-0934-7)  Did you exclude women for some reason?

5.- I suggest changing the title to something like this: Relationship between physical activity and quality of life in patients with colorectal cancer and postoperative defecation dysfunction

Reviewer 3 Report

Comments and Suggestions for Authors

The relevance of the article is beyond doubt. The positive impact of physical activity on human health has been studied by a large number of authors. However, in conditions of human health disorders, the factors of positive influence become even more important.

The purpose of the article has been achieved. The presentation of the scientific material allows us to fully reproduce the course of the study.

As for the comments:

line 59 - 64 characteristics of the methods of studying the level of physical activity should be presented in the research methods. The authors should pay more attention to the description of the problem of studying the level of motor activity, specifically in patients with defecation dysfunction.

In section 2.1. Study Design, Setting, and Patients should provide a full description of the study population, the total number of participants, age and gender distribution.

line 79 - 80 "excluding accidental bowel leakage" were included in the exclusion group?

line 103 - 106 avoid duplication of information Sexual interest was not calculated

how was the overall quality of life assessment conducted due to the lack of data on individual indicators

why only men's results are presented in Table 1

it is not clear from which number the percentage is calculated when characterising by gender (20 (66.7%) 22 (68.8%)).

in Table 2, to conditionally indicate a statistically significant difference between the two groups of respondents, for better perception of information.

Round 2

Reviewer 2 Report

Comments and Suggestions for Authors

1.- Please modify these lines located in material and methods section as follow please:

The participants were 65 adult patients of both sexes aged 20 years or older with postoperative defecation disorder due to colorectal cancer, who were divided into two groups: physically active (n=30) and physically inactive (n=32)

Author Response

Response to Reviewer 2 Comments

Dear Reviewer:

We wish to express our gratitude to the reviewer for their insightful comments. We have
attached our revised manuscript as well as a point-by-point response to the reviewers’
comments. Thank you in advance for your kind consideration.

Sincerely yours,
Hiromi Nakagawa
Gifu University Graduate School of Medicine, Gifu 501-1194, Japan
Tel.: 81-58-293-3248
[email protected]

Point 1: Please modify these lines located in material and methods section as follow please:
The participants were 65 adult patients of both sexes aged 20 years or older with postoperative
defecation disorder due to colorectal cancer, who were divided into two groups: physically
active (n=30) and physically inactive (n=32)

Response 1:
Thank you very much for reviewing our manuscript and offering valuable suggestions.
We have added the following text to lines 80-82:
" The participants were 65 adult patients of both sexes aged 20 years or older with
postoperative defecation disorder due to colorectal cancer, who were divided into two groups:
physically active (n=30) and physically inactive (n=32)"
